# *Hyptis obtusiflora* C. Presl ex Benth Methanolic Extract Exhibits Anti-Inflammatory and Anti-Gastritis Activities via Suppressing AKT/NF-κB Pathway

**DOI:** 10.3390/plants12051146

**Published:** 2023-03-02

**Authors:** Jieun Oh, Jae Youl Cho, Daewon Kim

**Affiliations:** 1Department of Integrative Biotechnology, Biomedical Institute for Convergence at SKKU (BICS), Sungkyunkwan University, Suwon 16419, Republic of Korea; 2Laboratory of Bio-Informatics, Department of Multimedia Engineering, Dankook University, Yongin 16890, Republic of Korea

**Keywords:** *Hyptis obtusiflora* C. Presl ex Benth, anti-inflammation, TLR4, AKT

## Abstract

Inflammation is an indispensable part of the human body’s self-defense mechanism against external stimuli. The interactions between Toll-like receptors and microbial components trigger the innate immune system via NF-κB signaling, which regulates the overall cell signaling including inflammatory responses and immune modulations. The anti-inflammatory effects of *Hyptis obtusiflora* C. Presl ex Benth, which has been used as a home remedy for gastrointestinal disorders and skin disease in rural areas of Latin America, have not yet been studied. Here, we investigate the medicinal properties of *Hyptis obtusiflora* C. Presl ex Benth methanol extract (Ho-ME) for inflammatory response suppression. Nitric oxide secretion in RAW264.7 cells triggered by TLR2, 3, or 4 agonists was reduced by Ho-ME. Reduction of inducible nitric oxide synthase (iNOS), cyclooxygenase (COX)-2, and interleukin (IL)-1b mRNA expression was observed. Decreased transcriptional activity in TRIF- and MyD88-overexpressing HEK293T cells was detected with a luciferase assay. Additionally, serially downregulated phosphorylation of kinase in the NF-κB pathway by Ho-ME was discovered in lipopolysaccharide-treated RAW264.7 cells. Together with the overexpression of its constructs, AKT was identified as a target protein of Ho-ME, and its binding domains were reaffirmed. Moreover, Ho-ME exerted gastroprotective effects in an acute gastritis mouse model generated by the administration of HCl and EtOH. In conclusion, Ho-ME downregulates inflammation via AKT targeting in the NF-κB pathway, and the combined results support *Hyptis obtusiflora* as a new candidate anti-inflammatory drug.

## 1. Introduction

Maintaining homeostasis is a lifetime objective for living organisms. Continuously interacting with the environment, organisms have evolved to protect themselves from extracellular materials such as bacteria, fungi, and viruses. Alien pathogens have unique glycan molecules [1], and target recognition contributes to the host’s immune system to distinguish “non-self” with pattern recognition receptors (PRRs). Immune receptors, including Toll-like receptors (TLRs) [2], C-type lectins [3], and Siglecs [4], are used to analyze pathogen-associated molecular patterns (PAMPs) and damage-associated molecular patterns (DAMPs). Of these receptors, TLRs discriminate the types of PAMP and activate inflammatory responses. TLRs are comprised of extracellular ligand-binding domains, transmembrane domains, and cytosolic Toll-interleukin-1 receptor (TIR) domains. In the eukaryotic system, 13 subtypes of TLRs with unique ligand patterns have been discovered [1]. While TLR1 and 2 or TLR2 and 6 form heterodimers and recognize Gram-positive bacteria, TLR4 can recognize Gram-negative endotoxins. Bacterial flagella are detected as positivity for TLR5, and ssRNA, dsRNA, and CpG DNA bind with TLR3, 7, 8, and 9. With TLR pattern recognition, adaptor proteins containing TIR domains, such as MyD88 and TRIF, are recruited toward the TIR domain of TLR. Intracellular signals then descend toward the nucleus to react to external stimuli by serial phosphorylation. Through phosphorylation, kinases are activated and amplify powerful signal transduction toward transcription factors. Finally, nuclear factor-κB (NF-κB) becomes phosphorylated and moves into the nucleus for the transcription of proinflammatory genes [5]. NF-κB orchestrates homeostasis via the rearrangement of cytokine expression [6,7,8,9]. However, unexpected, uncontrollable inflammation induced by NF-κB can cause a catastrophic situation. For example, *fortissimo* activation of NF-κB leads to a so-called “cytokine storm” [10], which can cause a health emergency. Moreover, prolonged inflammation can develop into chronic diseases [11]. Fine-tuning of NF-κB is important to harmonize homeostasis in the immune system.

According to previous reports, the Lamiaceae family is the sixth-largest plant family on earth. Approximately 258 genera and 7193 species are reported to belong to the Lamiaceae family [12]. Some of the most common species of Lamiaceae, such as classic oregano, sage, mint, basil, lemon balm, thyme, and rosemary, have been used as ornamental and medicinal plants [13]. Several studies have stated that the Lamiaceae family contains a variety of phytochemical compounds that have beneficial effects [14].

Among the hundreds of Lamiaceae genera, the *Hyptis* genus includes over 700 species of plants. Plants in the *Hyptis* genus contain several secondary metabolites including flavonoids [15], lactones [16], and steroids [17]. Plants from *Hyptis* have been commonly used to treat gastrointestinal problems, skin infections, and menstrual pain. According to ethnopharmacological literature, *Hyptis* plants are being researched for their pharmaceutical potential. Recently, Machado et al. have suggested that *Hyptis suaveolens* (L.) Poit has colon-protective effects [18], and Barbosa et al. have elucidated anti-inflammatory and antinociceptive effects of *Hyptis martiusii* Benth [19]. Additionally, anti-inflammatory bioactive metabolites in *Hyptis* plants have been identified [20].

*Hyptis obtusiflora* C. Presl ex Benth (Lamiaceae family and *Hyptis* genus) lives in Central America [21]. The ethnopharmacological literature reports oils infused with *Hyptis obtusiflora* as a folk medicine that treats gastrointestinal disorders and skin diseases [22]. Based on these reports, we hypothesized that *Hyptis obtusiflora* exerts anti-inflammatory effects, and we conducted a series of experiments to test the anti-inflammatory response efficacy and mechanism.

## 2. Results

### 2.1. Ho-ME Suppresses Nitric Oxide Production in TLR-Activated Macrophages

Nitric oxide (NO) assays were conducted to evaluate whether Ho-ME suppresses anti-inflammatory responses. RAW264.7 cells were inflamed by treatment with TLR4 ligand lipopolysaccharide (LPS), TLR2/TLR1 agonist Pam3CysSerLys4 (PAM3CSK), or TLR3 immunostimulant polyinosinic:polycytidylic acid (Poly I:C). TLR2-, TLR3-, and TLR4-mediated inflammation was effectively decreased when Ho-ME was co-administered with each pathogen. In Figure 1a, when 150 μg/mL of Ho-ME was administered for 24 h, NO production was inhibited to 14% of the control value. Moreover, only 12 h of Ho-ME treatment suppressed NO production by more than half compared to the positive control (Figure 1b). L-NAME, a standard NO-inhibitory compound, exhibited a dose-dependent reduction of NO production as expected (Figure 1c). Similar NO-reducing patterns were observed in the Pam3CSK-treated conditions (Figure 1d) and poly(I:C)-treated conditions (Figure 1e). To determine whether cell viability affects NO production, we employed MTT assays. Ho-ME rescued over 80% of cell viability compared to the non-treatment group at every concentration across both time points (Figure 1f,g).

In addition, time-of-flight mass spectrometry (TOF-MS) was performed to identify the main actors in the Ho-ME-mediated anti-inflammatory effects. Trifolin, genistin, and 4′,5,7,8-tetramethoxy-flavone were identified in the spectra (Figure 1h). Since other peaks are seen in the profile, detailed approaches like purification of individual components, analysis of chemical structures, and HPLC analysis with authentic compounds to identify each peak will be continued. For confirming whether these compounds can suppress NO production, the available compound (genistin) was exposed to RAW264.7 cells to evaluate the NO inhibition ability of Ho-ME. Treatment with only genistin not only dose-dependently suppressed NO production, but also downregulated NO production by up to 43% compared to that of the control group (Figure 1I). Currently, we are unable to check the NO inhibitory activity of other ingredients, due to no information about other peaks and the unavailability of these compounds. However, since it is important for us to know what compound can make synergistic or antagonistic activities between genistin and other compounds, we will continue additional work including the purification of ingredients and pharmacological tests with these compounds.

### 2.2. Ho-ME Altered Activities of Inflammation-Related Transcription Factors and mRNA Expression of Inflammatory Genes

NO, a paracrine signal messenger, stimulates the immune system as a chemoattractant that can induce vasodilation [23]. Several authors have recognized that NO production is controlled by inducible, neuronal, and endothelial NO synthases [23]. The experimental design of this paper assumed that mRNA expression of inducible NO synthase (iNOS) participates in NO production from RAW264.7 cells activated by TLR stimulators. Therefore, we first tested the level of iNOS under such conditions, as reported previously [24,25,26]. Moreover, IL-1β is a representative pro-inflammatory cytokine that can be regulated by NF-κB-dependent transcriptional activation [27,28,29]. Macrophage-dependent inflammation requires the expression of cyclooxygenase-2 (COX-2) to produce prostaglandin E_2_ (PGE_2_) [30,31,32]. Therefore, we examined whether these genes were suppressed by Ho-ME using both semi-quantitative reverse transcription polymerase chain reaction (RT-PCR, Figure 2a) and real-time quantitative RT-PCR (q-RT PCR, Figure 2b–d). The relative intensity of mRNA bands was reduced according to real-time q-RT PCR results. More specifically, 150 μg/mL of Ho-ME downregulated mRNA expression levels of iNOS and IL-1β to one-quarter that of the control group. These results imply that Ho-ME treatment could affect transcriptional activity. 

HEK293T cell viability assays (Figure 2g) were performed, followed by a luciferase assay to identify transcriptional activities (Figure 2e,f). Transcription factors NF-κB and AP-1 and adaptor molecules MyD88 and TRIF were used with β-galactosidase. Exogenous NF-κB activities decreased under Ho-ME treatment in both MyD88- or TRIF-transfected cells in a dose-dependent manner (Figure 2e,f). Moreover, transcription factor activities were confirmed with Western blotting. NF-κB heterodimer subunits p50 and p65 and each phosphorylated form were analyzed, as shown in Figure 2h. With Ho-ME treatment, phosphorylation of p50 and p65 was downregulated after 15–30 min.

### 2.3. Ho-ME Interrupts AKT Phosphorylation during an Intracellular Signaling Cascade

Considering the aim of the transduction of transcription factors in a signaling cascade, we investigated intracellular signals in the NF-κB pathway. We conducted a whole lysate assay to determine the proteins affected by Ho-ME. AKT was not affected by blocking IκBα and IKKα/β activation (Figure 3a), indicating AKT as a target of Ho-ME. To validate this, HA-tagged AKT1 and AKT2 constructs were introduced to HEK293T cells (Figure 3b,c). Both overexpressed constructs showed upregulated phosphorylation of the IKK complex, but Ho-ME changed the phosphorylation pattern by blocking the AKT series proteins (Figure 3b,c). A cellular thermal shift assay was adopted to elucidate the protein stability change caused by the interaction between Ho-ME and AKT. In this study, the thermo-dependent protein degradation environments were set to 44, 46, 48, 50, 52, 54, and 56 °C. Ho-ME treatment exhibited a considerable thermal stability shift at 46, 48, and 56 °C (Figure 3d). Additionally, to determine to which domain of AKT some of the active components of Ho-ME would bind, several AKT2 domain deletion mutants were transfected in HEK293T cells. As shown in previous research, the AKT structure is characterized by the kinase, pleckstrin homology (PH), and regulatory domains [33]. AKT2 truncated by the regulatory or PH domain exhibited similar downregulation of the level of p-IKKα/β to that of AKT wild-type when treated with Ho-ME, suggesting that Ho-ME physically interacts with the AKT kinase domain (Figure 3e). The effect of the AKT pathway inhibitor LY294002 was tested through an NO assay (Figure 3f).

### 2.4. Ho-ME Alleviated DAMP-Induced Acute Gastritis

The ability of Ho-ME to suppress inflammation and its mechanism was revealed through in vitro experiments. To ensure that Ho-ME would be effective in vivo, we investigated a gastritis model induced by HCl and ethanol. Stomach wall wounds were induced by oral administration of 60% EtOH/150 mM HCl (Figure 4a). Because the blood spot size is closely related to the severity of gastritis, the area of the blood spots was quantified with ImageJ software. As shown in Figure 4b, Ho-ME-administered groups showed smaller lesion areas than those of the ranitidine group, suggesting that Ho-ME attenuated DAMP-mediated inflammation. To confirm the molecular mechanism of Ho-ME in the mouse model, we analyzed the transcription levels of mRNA and proteins (Figure 4c–e). Ho-ME-mediated mRNA suppression was noted in patterns similar to the in vitro conditions, and p50 phosphorylation was downregulated as expected.

## 3. Discussion

The *Hyptis* genus has been widely used as traditional medicine for a variety of illnesses in tropical America. Due to its strong aromatic components and various phytochemical constituents, essential oils derived from the *Hyptis* genus have been studied for their effectiveness. However, only a few such species have been examined. *Hyptis obtusiflora* from plants in Central America was used as a home remedy. Literature from Peru states that *Hyptis obtusiflora* was used against ringworms and head wounds. In Ecuador, the use of *Hyptis obtusiflora* differed from province to province: using juice to heal wounds, infusions or ashes for a hot bath, and cooking leaves for skin infections and flu [22]. This practical knowledge was nearly lost until the Ecuadorian government conducted ethnopharmacological research on their native plants. Our study can not only help understand the medicinal properties of native plants, but also provide a link between traditional knowledge and future applications of these plants.

Inflammation involves a series of steps that protect the host from infection. Once patterns from pathogens are recognized via DAMP and PAMP, immune receptors activate the anti-inflammatory pathway. TLRs are responsible for mediating inflammatory responses. Regulating TLR responses is key to controlling inflammation because TLR acts like a toll gate that discriminates against pathogens [2]. TLRs sense changes in the extracellular environment and transduce signals to respond to external triggers. Dealing with environmental change is an urgent task for an organism, and signal transduction should be fast and accurate to support organism safety. Thus, sensitive control of early TLR response is very important. Careful modulation of TLR signaling is needed to maintain immune system equilibrium [2,5]. Rapid progression of inflammation, such as in sepsis, can threaten host safety [34]. Chronic inflammation can result in serious illnesses like cancer [35], inflammatory bowel disease [36], and rheumatoid arthritis [37]. In a series of NO assays, Ho-ME suppressed activation of not only TLR4, but also TLR2 and TLR3. Additionally, Ho-ME alleviated inflammation that was stimulated by DAMP. As seen in Figure 4, Ho-ME reduced mRNA expression of IL-1β and COX-2 and phosphorylation of p50 at the protein level. Taken together, these results show that Ho-ME might be an effective controller of front-line immune responses and could be a candidate for a universal anti-inflammatory drug.

In efficacy tests of herbal extracts, fragment analysis is a valuable process for understanding the material characteristics. Flavonoids, which are the main constituents of plant-based extracts, are secondary metabolites that have polyphenolic structures, and more than 5000 types of natural flavonoids have been reported [38]. In the present study, TOF-MS was used to identify fingerprints of Ho-ME. With this approach, genistin, an isoflavone that provides beneficial effects for general health [39], was found to be an ingredient of Ho-ME. In previous studies, it was reported that genistin inhibited cancer cell invasion and migration through the PI3K-AKT-mTOR axis and had a regulatory role in cell proliferation [40,41,42,43]. Cardioprotective effects achieved by blocking NF-κB pathways and suppressing proinflammatory cytokines [44] were reported as another beneficial effect of genistin. In addition, a galactose-conjugated flavonol, trifolin (kaempferol-3-O-galactoside), was detected in Ho-ME. Based on immunopharmacological approaches toward kaempferol and kaempferide, kaempferol has antimicrobial, anticancer, antiallergic, antidiabetic, antioxidant, and anti-inflammatory effects [45,46]. Trifolin exhibits both antifungal [47] and anticancer activities [48]. Moreover, Hataichanok et al. demonstrated anti-inflammatory effects of another identified compound in Ho-ME, 4′,5,6,7-tetramethoxy-flavone (scutellarein tetramethyl ether) [49]. This compound was reported to not only inhibit COX-2 and iNOS mRNA expression levels, but also suppress translocation of p65 under LPS-stimulation conditions. Taken together, these results led to the conclusion that these flavonoids in *Hyptis obtusiflora* play vital roles in negatively controlling NF-κB pathway-mediated inflammatory responses.

The ability to modulate IL-1β is necessary to maintain the consistency of the immune system. IL-1β is a precursor protein composed of 269 amino acids [50]. Once IL-1β is transcribed by NF-κB binding to the consensus binding sites of its promoter region, biologically active IL-1β is processed with caspase-1 and the inflammasome [51,52]. From a transcriptional point of view, Ho-ME effectively reduces NO production and proinflammatory gene expression. Especially, IL-1β mRNA expression is lower than that in the negative control group, as seen in Figure 2d and Figure 4d. However, excessive exposure to IL-1β can occur and exacerbate autoinflammatory diseases [53,54,55,56,57,58] and metabolic disorders [59,60,61,62]. High level of IL-1β promotes imbalanced immune states and increases in the Th17 cell population occur because of chronic exposure to IL-1β [63]. Moreover, at the hematopoietic cell level, IL-1β affects immune cell fate. Increased M1 polarizes macrophage and monocyte differentiation rates and activates B lymphocytes [64,65,66].

AKT, also known as protein kinase B, is a well-known key regulator within the NF-κB pathway. AKT exists as three subtypes, AKT1, AKT2, and AKT3. Even though the AKT isotypes have similar structures and biochemical characteristics, they have spatial differences in expression. AKT1 modulates cell survival and controls proliferation [67], AKT2 regulates insulin-mediated signaling [68], and AKT3 controls brain development [69]. AKT series proteins manage different aspects of cell biology, such as migration [70,71,72], ICAM-1 expression [73], respiratory burst [74], phagocytosis [75], and NF-κB signaling-related proteins [76]. In this study, an AKT2 domain mutant overexpression experiment was used to determine that Ho-ME binds with the AKT2 kinase domain (Figure 3). Despite the functional differences between the parts of the AKT series, they share similar sequences and structures. It can be deduced that Ho-ME can suppress AKT protein kinase activity by binding with the kinase domain. Therefore, Ho-ME has the potential as a candidate therapeutic for inflammatory and metabolic diseases by targeting AKT/NF-κB pathway, as summarized in Figure 5.

## 4. Materials and Methods

### 4.1. Materials and Reagents

International Biological Material Extract Bank (Daejeon, Korea) provided Ho-ME. RAW 264.7 and HEK293T cell lines were provided by ATCC (Rockville, MD, USA). Cell culture media (Roswell Park Memorial Institute 1640 (RPMI 1640), Dulbecco’s Modified Eagle Medium (DMEM), Opti-MEM, and streptomycin/penicillin) were purchased from Cytiva (Malborough, MA, USA). Fetal bovine serum (FBS) was obtained from Gibco (Grand Island, NY, USA). 1-Bromo-3-chloropropane, *N*^ω^-nitro-L-arginine methyl ester (L-NAME), 3-(4,5-dimethylthiazol-2-yl)-2,5-diphenyltetrazolium bromide (MTT), LPS (*E. coli* 0111:B4), and polyethylenimine (PEI) were purchased from Sigma Chemical Co. (St. Louis, MO, USA). TRI reagent^®^ was obtained from Molecular Research Center Inc. (Cincinnati, OH, USA). Primers used for PCR experiments were synthesized by Macrogen (Seongnam, Korea), and qPCRBIO SyGreen Mix Lo-ROX and HS Taq PreMix Red were purchased from PCR Biosystems (London, United Kingdom). Antibodies specific for β-actin and p-p50 were purchased from Santa Cruz Biotechnology (Dallas, TX, USA), and both the total and phosphorylated forms of AKT series, IKKα/β, IκBα, HA-tag, p65, and p50 were acquired from Cell Signaling Technology (Danvers, MA, USA). Ethanol, methanol, isopropanol, and hydrochloric acid were manufactured by Daejung Chemicals and Metals (Seoul, Korea).

### 4.2. Plant Extract Processing

The leaves of *Hyptis obtusiflora* (53 g) were extracted in 1 L of 99.9% (*v*/*v*) methanol by repeated sonication (15 min) and rest (2 h) for 3 days at 45 °C. Filtration and concentration of *Hyptis obtusiflora* were conducted as reported previously [77]. A total of 1.3 g of *Hyptis obtusiflora* methanol extract powder was obtained by lyophilization.

### 4.3. Cell Culture

RAW264.7 cells were maintained in a RPMI 1640 medium. HEK293T cells were cultured in DMEM. Both media contained 10% heat-inactivated FBS, penicillin (100 U/mL), and streptomycin (100 μg/mL). For subculture, RAW264.7 cells were detached with a scraper, and HEK293T cells were treated with 1 mL of trypsin (Cytiva, Malborough, MA, USA) for detachment. All cells were subcultured every 3 days, and only cells younger than passage 40 were used for in vitro experiments.

### 4.4. NO and MTT Assays

RAW264.7 cells and HEK293T cells were seeded in 96-well plates and used in experiments after overnight incubation. For the NO assay, Ho-ME was administered at the indicated final concentration (0, 25, 50, 75, 100, or 150 μg/mL), and LPS was added. After 12 or 24 h, supernatants and Griess reagent were mixed at a 1:1 ratio. Absorbance at 540 nm was detected via a spectrometer. Cell viability was measured after Ho-ME treatment and incubation for 24 h. After the addition of 10 μL of MTT solution and 4 h of incubation, 100 μL of MTT stop solution was added to each well. After overnight incubation to dissolve the purple-colored formazan, the absorbance at 570 nm was detected with a spectrometer.

### 4.5. RNA Extraction and Polymerase Chain Reaction

RAW264.7 cells were seeded in 6-well plates and cultured overnight at 37 °C in a 5% CO_2_ incubator. Ho-ME was added 30 min before LPS stimulus. After 6 h of incubation, all supernatants were aspirated, and the cells were harvested. RNA was isolated with the Trizol and bromochloropropane method, as previously reported. cDNA was synthesized according to the manufacturer’s instructions. Both RT-PCR and real-time PCR were conducted using 100 ng of cDNA. The used primer sequences are listed in Table 1. In real-time PCR, all mRNA levels were expressed relative to the level of GAPDH expression.

### 4.6. Luciferase Assay

HEK293T cells were plated in 24-well plates at a density of 1 × 10^6^ cells/mL. Cells were transfected with the following plasmids: one with a luciferase gene-included transcription factor (NF-κB-Luc) and one with an adaptor (FLAG-MyD88 or CFP-TRIF) and β-galactosidase. After 24 h of transfection, the HEK293T cells were treated with Ho-ME or vehicle and incubated for another 24 h. The supernatants were aspirated, and cells were lysed with 400 μL of luciferase lysis buffer. Luminescence detection was performed according to a previously reported luciferase assay system [78]. Each luminescence reading was normalized to that of β-galactosidase.

### 4.7. Western Blot

Cells or tissues were lysed with RIPA buffer (20 mM Tris-HCl, pH 7.4, 2 mM ethylene glycol tetraacetic acid, 50 mM β-glycerol phosphate, 1 mM sodium orthovandate, 1 mM dithiothreitol, 1% Triton X-100, 10% glycerol, 10 μg/mL aprotinin, 10 μg/mL pepstatin, 1 mM benzamide, and 2 mM phenylmethylsulfonyl fluoride [PMSF]). After lysis, supernatants were collected via centrifugation. Proteins were quantified using a Bradford assay. Each experimental sample contained 20 μg/mL of total protein concentration. SDS-PAGE was conducted with 20 μL samples, and electrophoresis was performed at 100 V. Proteins in gels were transferred to a PVDF membrane at 100 V for 2 h. Membranes were soaked in 3% bovine serum albumin (BSA) in 1 × TBST solution for 1 h to block non-specific binding. Primary antibodies were diluted with a 3% BSA solution at a 1:2500 ratio and were incubated for 3–4 h at room temperature. The following specific primary antibodies were used in this study: AKT (#9272), p-AKT (Ser473) (#4058), IKKα (#2682), p-IKKα/β (#2697), IκBα (#9242), p-IκBα (#9246), p50 (#12540), p65 (#8242), p-p65 (#3039), HA-tag (#2367) (Cell Signaling Technology), p-p50 (SC271908), and β-actin (SC4778) (Santa Cruz Biotechnology). After three washes with 1 × TBST solution, secondary antibodies were incubated at a 1:2500 ratio for 2 h at room temperature. When the incubation ended, the membranes were washed with 1 × TBST three times to remove unintended secondary antibody binding. ECL-mediated chemiluminescence was detected with a ChemiDoc system (BIO-RAD, Hercules, CA, USA).

### 4.8. Overexpression

HEK293T cells were seeded at a confluency of 1 × 10^6^ cells/mL in six-well plates. For overexpression, 8 μg of each construct was transfected with PEI. After 24 h of transfection, the old media was aspirated, and Ho-ME or vehicle was administered for another 24 h. Harvested cells were lysed with RIPA buffer. After protein quantification and mixing with loading buffer, phosphorylation of signaling cascade proteins was analyzed with Western blotting.

### 4.9. Cellular Thermal Shift Assay

HEK293T cells were transfected with HA-AKT2 plasmid for 24 h and then treated with Ho-ME or vehicle for another 24 h. Harvested cells were placed in PCR tubes and heated for 3 min at each intended temperature and then chilled at room temperature for 3 min. Cells were placed in a deep freezer and subjected to three thaw–freeze cycles with liquid nitrogen. Finally, lysates were centrifuged to collect the supernatant. After mixing with a loading buffer, the lysates were subjected to Western blot, as previously reported [79].

### 4.10. Mice

A collection of 25 male, five-week-old ICR mice was purchased from Orient Bio (Sungnam, Korea). The animals had free access to food and water ad libitum during a one-week acclimation. All mice were housed in autoclaved plastic cages. A maximum of five mice were housed in each cage. A 12 h:12 h light–dark cycle was applied to maintain the homeostasis of the circadian rhythm. Animal care and experimental procedures were performed in accordance with the guidelines for animal care (NIH Publication 80-23, revised in 1996) and were approved by the Institutional Animal Care and Use Committee of Sungkyunkwan University (approval number: SKKUIACUC2020-06-30-1).

### 4.11. Acute Gastritis Mouse Model Generated with HCl/EtOH

The mice were randomly split into five groups: Normal, Control, Ho-ME-treated (100 and 150 mg/kg), and Ranitidine-treated (40 mg/kg). A 0.5% solution of carboxy methyl cellulose (CMC) was used as the vehicle, and the normal and control groups received only 0.5% of CMC solution. Every drug was delivered orally, and 100 μL of solution was administered for each injection. During the experiments, mice were not allowed to be fed but had access to water. Oral administrations were conducted thrice in total. On the first day of experiments, oral injections were performed twice in 8 h. The next day, 5 h after final drug administration, 300 μL of 60% EtOH/150 mM HCl was delivered to each mouse except those in the normal group. After 1 h, mice were anaesthetized, and tissues were harvested. Obtained samples were photographed for blood spot analysis and then stored at -80 °C. Blood spot areas were quantified with ImageJ software, and the stomachs were analyzed via PCR and Western blot assays. Stomach grinding in liquid nitrogen was performed before RNA extraction or lysis with RIPA buffer.

### 4.12. Quadrupole Time-of-Flight LC/MS

The phytochemical characteristics of Ho-ME were confirmed with a Xevo G2-XS quadrupole time-of-flight liquid chromatography/mass spectrometry (Q-TOF-LC/MS) system (Waters, Milford, MA, USA). Reverse-phase BEH C18, 2.1 × 100 mm, 1.7 mm (Waters) resin was packed for UPLC, with 0.1% formic acid in water and acetonitrile as mobile phases (A) and (B), respectively. The gradient method was used as previously reported [77]. The designated column temperature was 45 ℃, and the mobile phase flow rate was 0.3 mL/min. A 2 μL sample of Ho-ME was injected for each experiment. The mass spectrometry conditions were set as described previously [80]. All data were acquired and analyzed with Waters LC-MS-QTOF MassLynx Software version 4.2 and Waters UNIFI Portal Software (Waters).

### 4.13. Statistical Analysis

The data are presented as the mean and standard deviation of independent replicate experiments, six for NO assay and MTT assay, five for luciferase assay and in vivo acute gastritis model, and 4 technical replicates per group in real-time qRT-PCR tests. Statistical comparisons were performed with Student’s *t*-test, the Mann–Whitney *U* test, or one-way analysis of variance (ANOVA). A *p*-value < 0.05 was considered statistically significant. All statistical analyses were performed using GraphPad Prism 8 software (GraphPad, San Diego, CA, USA).

## 5. Conclusions

This study suggests that Ho-ME has an inflammation-suppressive ability by targeting AKT kinase in pathogen-stimulated murine macrophages, as summarized in Figure 5. Ho-ME suppresses mRNA expression of proinflammatory cytokines and phosphorylation of inflammatory response-related proteins, and transcription factor activities are also down-regulated. Moreover, the mechanism of Ho-ME-mediated inflammation alleviation is reaffirmed in an in vivo model. These results demonstrate the potential of Ho-ME as a pharmaceutical agent against a variety of inflammatory diseases.

## Figures and Tables

**Figure 1 plants-12-01146-f001:**
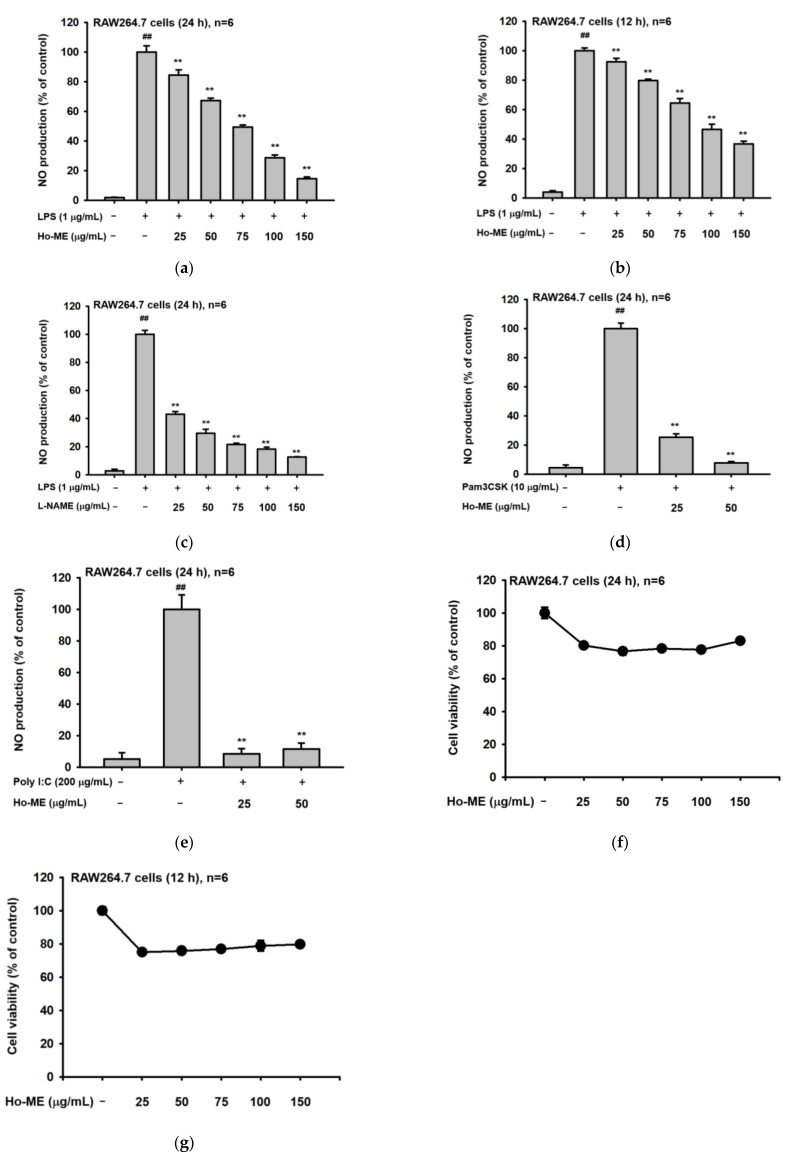
Ho-ME alleviated NO production in activated macrophages. (**a**,**b**) NO production level was assessed according to Ho-ME treatment in RAW264.7 cells stimulated by LPS for 24 (**a**) and 12 h (**b**). (**c**) The ability of L-NAME, a NOS inhibitor, to suppress the production of NO in LPS-treated RAW264.7 cells was tested by NO production assay. (**d**,**e**) NO inhibitory level of Ho-ME was observed in (**d**) PAM3CSK4- and (**e**) Poly I:C-treated RAW264.7 cells. (**f**,**g**) Cell viability of RAW 264.7 cells on Ho-ME treatment was assessed by MTT assay. (**h**) Phytochemical fingerprinting profile of Ho-ME was obtained by liquid chromatography-quadrupole time-of-flight mass spectrometric (LC-QTOF/MS) analysis. (**i**) Suppression of NO by genistin treatment was confirmed by NO assay under the same conditions. All data are presented as mean ± standard deviation (SD) calculated from the indicated number of independent samples. ##: *p* < 0.01 compared to the non-stimulated group, and **: *p* < 0.01 compared to the control groups.

**Figure 2 plants-12-01146-f002:**
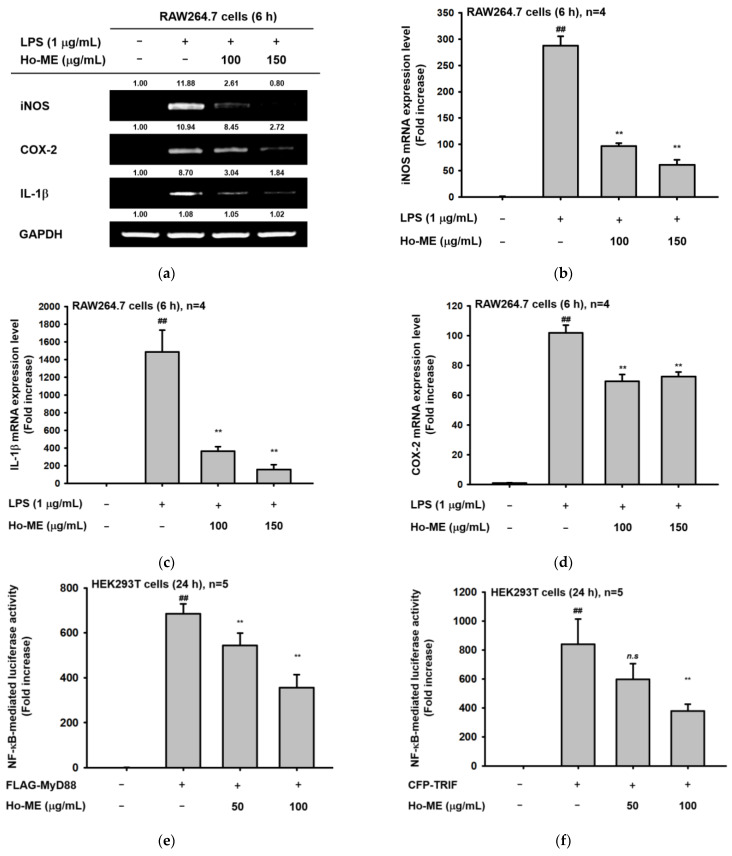
Ho-ME altered transcriptional activities along the NF-κB pathway. (**a**) Semi-quantitative PCR results of proinflammatory cytokine- and NO production-related genes. (**b**–**d**) The mRNA expression level of each gene was quantified by qRT-PCR. (**e**,**f**) MyD88- or TRIF-activated NF-κB activities were analyzed via a luciferase assay in HEK293T cells. Transcriptional activities were calculated by luminescence. (**g**) Effect of Ho-ME on HEK293T cell viability. (**h**) After Ho-ME treatment, phosphorylation level changes of NF-κB subunits were detected with Western blot. The numbers above each cell in (**a**) and (**h**) are relative intensity values, measured using ImageJ software (Wayne Rasband, NIH, Bethesda, MD, USA). ##: *p* < 0.01 compared to the non-stimulated group and **: *p* < 0.01 compared to the control groups.

**Figure 3 plants-12-01146-f003:**
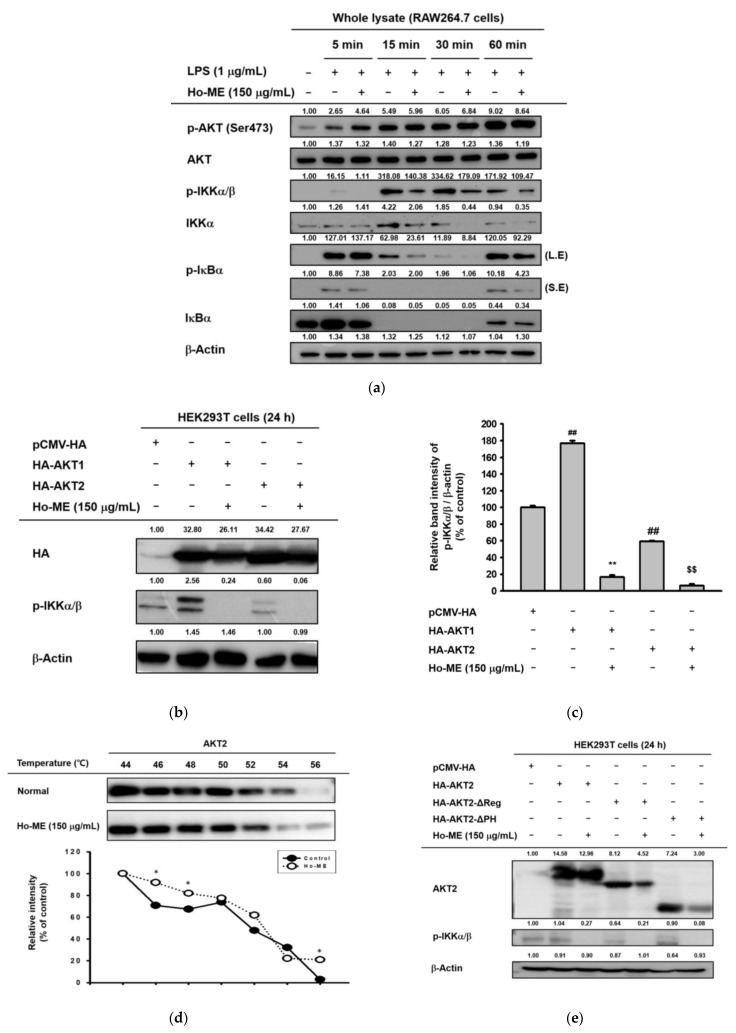
Ho-ME affected intracellular signaling by targeting AKT. (**a**) Western blot image of inflammatory response-related proteins. (**b**,**c**) The protein target of Ho-ME was reaffirmed in AKT1- and AKT2-overexpressing HEK293T cells. p-IKKα/β was determined by immunoblotting analysis. The relative band intensity level of p-IKKα/β was calculated with ImageJ software. (**d**) Ho-ME enhanced the thermostability of AKT2 by binding. (**e**) The Ho-ME binding domain was elucidated with domain-truncated AKT2 overexpression. (**f**) LY294002, a PI3K inhibitor, eliminated NO production in RAW264.7 cells. All data are presented as mean ± standard deviation (SD) calculated from the indicated number of independent samples. The numbers above each image in (**a**,**b**) and (**d**,**e**) are relative intensity values, measured using ImageJ software. L.E.: Long exposure. S.E.: Short exposure. ##: *p* < 0.01 compared to the non-stimulated group, *: *p* < 0.05 and **: *p* < 0.01 compared to the control or AKT1 overexpression group, and $$: *p* < 0.01 compared to AKT2 overexpression group.

**Figure 4 plants-12-01146-f004:**
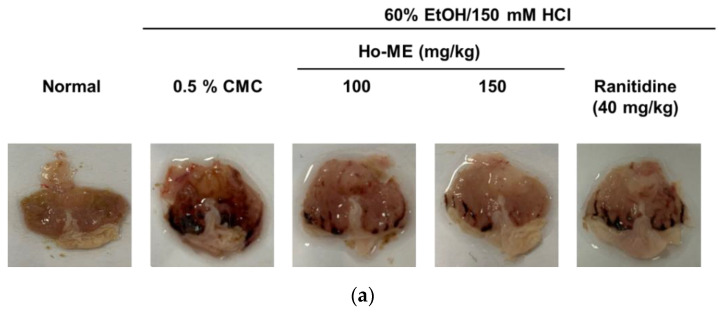
The gastroprotective ability of Ho-ME in an in vivo gastritis model. (**a**) Photographs of the stomach at 1 h after administration of mixed HCl and EtOH. (**b**) Blood spot areas were quantified with ImageJ software. (**c**–**e**) Inflammation-related mRNA expression levels were analyzed in the gastric tissue. (**f**) After Ho-ME treatment, downregulation of transcription factor phosphorylation was detected. The numbers above the images are the relative intensity values measured with ImageJ software. All data are presented as mean ± standard deviation (SD) calculated from the indicated number of independent samples. ##: *p* < 0.01 compared to the non-stimulated group and *: *p* < 0.05 and **: *p* < 0.01 compared to the control group.

**Figure 5 plants-12-01146-f005:**
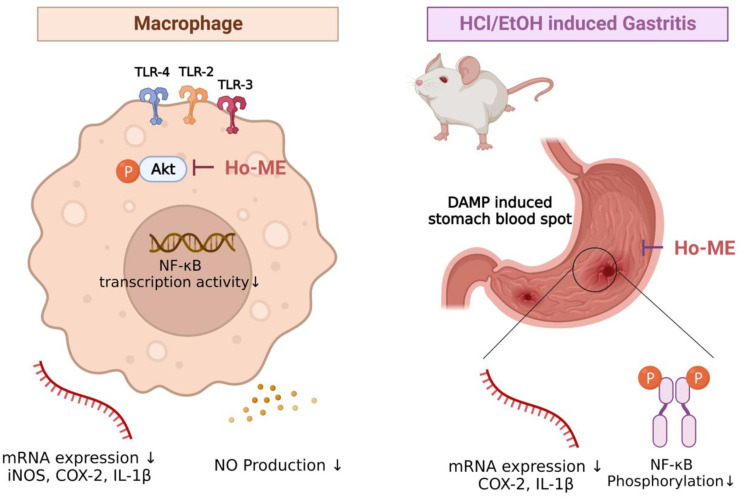
Putative scheme of AKT-targeted anti-inflammatory response by Ho-ME at the molecular, cellular, and in vivo levels.

**Table 1 plants-12-01146-t001:** PCR primer sequence list.

PCR	Organism	Target	Direction	Sequence (5′ to 3′)
Semi-quantitative	Mouse	IL-1β	Forward	CAGGATGAGGACATGAGCACC
Semi-quantitative	Mouse	IL-1β	Reverse	CTCTGCAGACTCAAACTCCAC
Semi-quantitative	Mouse	iNOS	Forward	TGCCAGGGTCACAACTTTACA
Semi-quantitative	Mouse	iNOS	Reverse	ACCCCAAGCAAGACTTGGAC
Semi-quantitative	Mouse	COX-2	Forward	TCACGTGGAGTCCGCTTTAC
Semi-quantitative	Mouse	COX-2	Reverse	CTTCGCAGGAAGGGGATGTT
Semi-quantitative	Mouse	GAPDH	Forward	CACTCACGGCAAATTCAACGGCA
Semi-quantitative	Mouse	GAPDH	Reverse	GACTCCACGACATACTCAGCAC
Realtime	Mouse	IL-1β	Forward	GTGAAATGCCACCTTTTACAGTG
Realtime	Mouse	IL-1β	Reverse	CCTGCCTGAAGCTCTTGTTG
Realtime	Mouse	iNOS	Forward	GCCACCAACAATGGCAACAT
Realtime	Mouse	iNOS	Reverse	TCGATGCACAACTGGGTGAA
Realtime	Mouse	COX-2	Forward	TTGGAGGCGAAGTGGGTTTT
Realtime	Mouse	COX-2	Reverse	TGGCTGTTTTGGTAGGCTGT
Realtime	Mouse	GAPDH	Forward	TGTGAACGGATTTGGCCGTA
Realtime	Mouse	GAPDH	Reverse	ACTGTGCCGTTGAATTTGCC

## Data Availability

The data used to support the findings of this study are available from the corresponding author upon request.

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
