# Peer review of "Hyptis obtusiflora C. Presl ex Benth Methanolic Extract Exhibits Anti-Inflammatory and Anti-Gastritis Activities via Suppressing AKT/NF-κB Pathway"

_plants, 2023, doi:10.3390/plants12051146_

Round 1

Reviewer 1 Report (Previous Reviewer 2)

The topic of the manuscript "Hyptis obtusiflora C.Presl ex Benth methanolic extract exhibits anti-inflammatory and anti-gastritis activities via suppressing AKT/NF-kB pathway" is interesting, and the manuscript is well organized and presented. But the authors need to revise the manuscript.

1. in Fig. 2b, d, why n=2, if n=2, how to get the SD? please check.

Author Response

Reviewer #1

The topic of the manuscript "Hyptis obtusiflora C.Presl ex Benth methanolic extract exhibits anti-inflammatory and anti-gastritis activities via suppressing AKT/NF-kB pathway" is interesting, and the manuscript is well organized and presented. But the authors need to revise the manuscript.

1. in Fig. 2b, d, why n=2, if n=2, how to get the SD? please check.

*Answer: thank you reviewer. We have conducted real-time PCR work with quadruplicate loading and analyzed statistics based on result again. We also have similar data with different experiment done independently. You can check figure 2B, C, and D on manuscript page 3 and L434.

Reviewer 2 Report (Previous Reviewer 1)

The overall quality of manuscript was markedly improved. 

The caption of figure 5 should be self-explanatory and referred somewhere in the results or discussion rather than conclusions.

Author Response

Reviewer #2

The overall quality of manuscript was markedly improved. 

The caption of figure 5 should be self-explanatory and referred somewhere in the results or discussion rather than conclusions.

**Answer: Dear reviewer, we thank for your suggestion. We changed caption as you pointed out. Please kindly find at L447-448. We also referred figure 5 in Discussion section (see L290-291).

Reviewer 3 Report (New Reviewer)

The manuscript entiled "Hyptis obtusiflora C.Presl ex Benth methanolic extract exhibits anti-inflammatory and anti-gastritis activities via suppressing AKT/NF-B pathway" discusses the use of the alcoholic extract as ani-inflammatory agent. Generally, the manuscript is well written however several points needed to be justified.

1- The provided figure 1I is not enough to identify genistin, trifolin and 4`,5`,7`,8`-tetramethoxyflavone and no comparison with authentic individuals. even though these compounds are not the major components according to the provided figure, it is not clear.

2- No mass spectra charts provided.

3- Is there any synergism or antagonism between genstin and other extract's components.

4-In the conclusion section L435-438, this deduction is not supported and only can be proposed for future work.

Round 2

Reviewer 3 Report (New Reviewer)

The manuscript is ready for publication in Plants.

This manuscript is a resubmission of an earlier submission. The following is a list of the peer review reports and author responses from that submission.

Round 1

Reviewer 1 Report

The manuscript is focused on the study of anti-inflammatory effects of Hyptis obtusiflora methanol extract. Based on the journal choice I would expect more emphasize on the metabolic analysis of plant extract. However, only a passing mention was made about potential metabolites involved in the anti-inflammatory response. 

The manuscript requires an extensive language correction, because there are many inappropriate words/phrases, such as "Hyptis obtusiflora is from the Lamiaceae family, Hyptis genus and lives in Central America"; the language style consists of a combination of author plural and past participle. 

The short survey on the anti-inflammatory effects of genus Hyptis representatives is absolutely missing in both introduction and discussion. There are several relevant paper that should be cited (Machado et al. 2021, Barbosa et al. 2022, ...). 

Many subsection headers should be improved, such as "Cell culture". 

The selection of investigated genes (COX-2 and iNOS) for qRT-PCR is not clear. Why did the authors choose these genes for expression analysis? The conclusion that "Ho-ME treatment affects gene transcription" requires better evidence than the expression analysis of severeal selected genes. 

In general, the results are written rather descriptively. I would recommend to improve the core of the manuscript. The expression "main flavonoid component" is a little confusing. Did the authors mean the most produced flavonoid compound? 

The effect on the cells and mouse model was tested using the whole extract, so the linking of  anti-inflammatory activities with secondary metabolites was hypothetized solely based on the literature.

Reviewer 2 Report

The topic of the manuscript entitled "AKT-mediated anti-inflammatory response to a methanolic extract of Hyptis obtusiflora C.Presl ex Benth" is interesting, and the manuscript is well written and organized. But the manuscript needs some minor revisions before publication.

1. please add significant letter in Fig.1 a,b;

2. please add n=? in the corresponding Tables and Figures;

3. please check the reference format, esp. to those journal name;

4. Others comments were marked in the text.
